# Computerized Cognitive Rehabilitation Training for Ugandan Seniors Living with HIV: A Validation Study

**DOI:** 10.3390/jcm9072137

**Published:** 2020-07-07

**Authors:** Amara E. Ezeamama, Alla Sikorskii, Parvathy R. Sankar, Noeline Nakasujja, Michael Ssonko, Norbert E. Kaminski, David Guwatudde, Michael J. Boivin, Bruno Giordani

**Affiliations:** 1Department of Psychiatry, College of Osteopathic Medicine, Michigan State University, East Lansing, MI 48824, USA; sikorska@msu.edu (A.S.); sankarpa@msu.edu (P.R.S.); boivin@msu.edu (M.J.B.); 2Department of Psychiatry, Makerere University College of Health Sciences, Kampala P.O. Box 7062, Uganda; drnoeline@yahoo.com; 3Mildmay Hospital, Kampala P.O. Box 24985, Uganda; mikssonko@gmail.com; 4Department of Pharmacology and Toxicology and Institute for Integrative Toxicology, Michigan State University, East Lansing, MI 48824, USA; kamins11@msu.edu; 5Department of Epidemiology and Biostatistics, School of Public Health, Makerere University College of Health Sciences, Kampala P.O. Box 7072, Uganda; dguwatudde@musph.ac.ug; 6Department of Neurology & Ophthalmology, Michigan State University, East Lansing, MI 48824, USA; 7Department of Psychiatry, University of Michigan, Ann Arbor, MI 48109, USA; 8Departments of Psychiatry, Neurology and Psychology, School of Nursing, University of Michigan, Ann Arbor, MI 48109, USA; giordani@umich.edu

**Keywords:** computerized cognitive rehabilitation therapy, neurocognitive impairment, quality of life, frailty, HIV/AIDS, aging

## Abstract

The feasibility, acceptability and preliminary efficacy of computerized cognitive rehabilitation therapy (CCRT) for mitigating neurocognitive decline was evaluated in African adults ≥50 years old. Eighty-one Ugandans with (*n* = 40) and without (*n* = 41) chronic human immunodeficiency viruses (HIV) were allocated CCRT—i.e., 20–45-min cognitive training sessions with culturally adapted video games delivered via Captain’s Log Software, or standard of care (SOC). Pre and post (i.e., 8-weeks later) intervention performance based neurocognitive tests, quality of life (QOL) and frailty related phenotype (FRP) were determined in all respondents. Multivariable linear regression estimated CCRT- vs. SOC-related differences (β) in neurocognitive batteries, QOL and FRP. Effect sizes (ES) for estimated β were calculated. CCRT protocol was completed by 92.8% of persons allocated to it. Regardless of HIV status, CCRT was associated with higher performance in learning tests than SOC—interference list (β = 1.00, 95%CI: (0.02, 1.98); ES = 0.43) and delayed recall (β = 1.04, 95%CI: (0.06, 2.02); ES = 0.47). CCRT effect on verbal fluency was clinically important (ES = 0.38), but statistical significance was not reached (β = 1.25, 95%CI: (−0.09, 2.58)). Among HIV-positive adults, clinically important post-CCRT improvements were noted for immediate recall (ES = 0.69), working memory (ES = 0.51), verbal fluency (ES = 0.51), and timed gait (ES = −0.44) tasks. Among HIV-negative adults, CCRT resulted in moderate post-intervention improvement in learning tests (ES = 0.45) and large decline in FRP (ES = −0.71), without a positive effect on simple attention and visuomotor coordination tasks. CCRT intervention is feasible among older Ugandan adults with potential benefit for learning and verbal fluency tests regardless of HIV status and lowering FRP in HIV-negative older adults.

## 1. Introduction

Demographic and epidemiologic transitions have expanded the population of older individuals, [1] and neurocognitive dysfunction/impairment has become a public health problem worldwide. These transitions combine with forces of globalization and changes in nutrition to increase the risk of non-communicable diseases (NCDs) [2] such as type II diabetes, hypertension and cardiovascular diseases [3] in many Sub-Saharan African countries. In Uganda, the rise in NCD in the general population has been characterized as severe [4]. NCDs are often accompanied by metabolic dysregulations and frailty, i.e., vulnerability due to decline in function across multiple physiologic systems, which constitute independent risk factors for age-related neurocognitive dysfunction, disability and low quality of life (QOL) [4,5,6,7].

Age-related changes are seen in executive activities involving working memory, attention, task switching, and motor response speed [8]. These cognitive/behavioral changes with aging are matched with neuroanatomical changes, including reduced volume of gray matter in the brain; progressive thinning out of the cerebral cortex, particularly in the frontal lobes; lower levels of dopamine and serotonin neurotransmitters; and lower binding potential of neurotransmitters to respective receptors [8]. Older adults living with human immunodeficiency viruses (HIV) infection are at especially high risk of premature neurocognitive decline, frailty and low QOL. Aging and NCD-related risk factors synergize with virus-mediated neuroinflammation [9] and HIV-treatment-related side-effects such heart disease [10] to advance the time to onset of various aspects of aging-related neurocognitive dysfunction, frailty and low QOL [11]. One functional consequence of these outcomes is that complex tasks, such as cooking, are the first areas affected, followed by the ability to execute instrumental activities of daily living, as neurocognitive decline and frailty worsen [11].

There are no known effective treatments to mitigate aging or HIV-associated premature neurocognitive decline. However, it has been shown that when the brain is exposed to challenging and stimulating tasks, it will grow (like neurogenesis in hippocampus), while deprivation results in shrinkage of the brain regardless of age [12]. Hence, tasks that incorporate elements of newness and challenge have the potential to maintain or improve neurocognitive performance and increase QOL in vulnerable populations [12]. Recent research suggests that cognitive stimulation programs in the form of different activities that are geared towards memory, attention, gnosis, and executive functions, when completed for 24 weeks by HIV affected individuals, result in better neuropsychological test performance, even when adjusted for age, gender, education, and race [10]. Our group [13,14] and others [10,15,16,17] have adapted CCRT programs for computer delivery in the form of simple game-like programs that can be used by lay individuals on a laptop or tablet with minimal training. This potential utility of computerized cognitive rehabilitation therapy (CCRT) for improving aspects of neurocognitive function recently has been demonstrated for African children with cerebral malaria [18,19,20,21] and in several Western studies, including older adults enrolled in the multi-center trial, Advanced Cognitive Training for Independent and Vital Elderly (ACTIVE) trial [22], adults with heart disease [23,24], and adults subjected to prolonged bed-rest [13,14]. However, the feasibility of CCRT intervention for mitigating aging and HIV-associated neurocognitive decline in the African setting is unknown. Hence, in this study, we evaluated feasibility of a culturally adapted CCRT intervention among older Ugandans with and without HIV-infection. Further, this study assesses the preliminary efficacy of a CCRT intervention via effect sizes for improvement in neurocognitive outcomes (primary outcome), QOL and reduction in frailty-related phenotype (secondary outcomes) among older adults with and without chronic HIV infection. We further explore the variations in CCRT feasibility and benefit by HIV status.

## 2. Experimental Section

### 2.1. Study Design

This is a non-randomized trial of CCRT vs. standard of care (SOC, i.e., no CCRT) in older HIV-positive patients connected to HIV care at Mildmay Uganda Hospital and HIV-negative community controls who were matched to HIV-positive adults by age (±5 years), sex and village of residence. Consenting eligible participants were allocated on a first-come-first-served basis to CCRT intervention or SOC in alternating blocks of eight participants of same HIV status till target sample size was attained.

### 2.2. Setting

The feasibility trial was conducted at Mildmay Uganda Hospital—a tertiary comprehensive health center that provides care and treatment for HIV-infected persons as its primary function. At the time of study initiation, Mildmay Clinic provided care for an estimated 15,300 HIV-infected adults, of whom 15,200 were on combination antiretroviral therapy (ART).

### 2.3. Recruitment and Enrolment Process

Participants were enrolled between July 2017 and April 2018. HIV-infected adults were recruited from the study hospital and matched to HIV-negative community controls by age, sex and village of residence. Due to staffing and space constraints combined with the need to preserve confidentially of HIV-positive individuals, allocation to cognitive training was done in blocks by HIV status. A maximum of eight individuals could be cognitively trained at any one time. Hence, the first 16 individuals were identified and the first eight that consented to multiple study visits for the purpose of implementing the CCRT protocol were allocated to CCRT arm. The remaining eight individuals were allocated to SOC arm. The second block of enrollees were demographically matched HIV-negative controls allocated to either CCRT or SOC in the same manner as described for HIV-positive adults.

### 2.4. Eligibility and Exclusion Criteria

Eligibility criteria included: (a) men or women aged ≥50 years old, (b) if HIV+ must be connected to HIV care at the Mildmay Uganda Hospital, (c) must reside within 25 km of the hospital, and (d) if HIV-negative, must consent to HIV-testing at enrolment. All participants provided written informed consent for study participation and were willing to return for implementation of the CCRT protocol if allocated CCRT. Subjects who were very ill or physically disabled and thus unable to undergo performance-based assessments were excluded.

### 2.5. Intervention Allocation and Standard of Care (SOC)

Participants in the CCRT intervention arm received 20–45-min sessions of cognitive training on laptop computers with culturally adapted video games through Captain’s Log Software delivered 2 days per week (two training sessions per day) over a minimum of 5 weeks. All participants wore sound-deadening headphones and sat far enough apart from one another to avoid being distracted by other participants training at the same time. Research assistants proficient in the local language, Luganda, supervised CCRT and provided instructions or guidance as needed. CCRT intervention consisted of two components—culturally adapted Brain Powered Games—Uganda (BPG-U, 20 min) and Spatial Navigation Training (SNT, 25 min). Participants in the SOC arm were not cognitively trained, but had neuropsychological, QOL and frailty tests on the same schedule as the CCRT group.

At all times during this trial, HIV-positive participants continued to receive standard medical care according to Uganda’s Ministry of Health (MoH) guidelines. In brief, HIV-positive adults are initially assigned to the first line three drug combination highly active antiretroviral therapy. Typically, the first line cocktail includes: Abacavir or zidovudine (AZT) or tenofovir (TDF) + lamivudine (3TC) + efavirenz (EFV) or nevirapine (NVP). In case of treatment failure or drug toxicity, patients that were on abacavir or TDF are assigned to AZT + 3TC plus a protease inhibitor which could be Atazanavir or lopinaviar boosted with ritonavir. All HIV+ persons are maintained on cotrimoxazole prophylaxis or dapsone for life and strongly encouraged to consistently sleep under insecticide-treated mosquito nets to limit the risk of malaria. In the absence of a system of preventive primary health care in Uganda, most “healthy” HIV-negative individuals are not connected to routine care unless they are ill. Thus, usual study-related medical care for HIV-negative participants included referral to the national referral hospital for further management in case they develop clinical signs/symptoms. This is consistent with the standard practice in the Uganda health system.

### 2.6. Measurements

At baseline only, we administered a detailed background information questionnaire to obtain socio-demographic data, including age, parity, education, residence, employment status, socio-economic status variables, marital status, and past/current use of tobacco, alcohol, and other substances. The Center for Epidemiological Studies-Depression scale was used to assess depressive symptoms [25]. Psychosocial adversity was measured as acute (within 30 days), recent (within 5 years) and as a lifetime cumulative adversity, using the perceived stress scale and recent and lifetime adversity questionnaires adapted and translated for the study setting [26]. Blood samples were collected from each participant at enrollment to measure CD4 counts, complete blood counts (CBCs), bilirubin, creatinine, and alanine transferase, as part of clinical standard of care for all patients. In addition, lipids profile testing (high density lipoproteins (HDL), low density lipoproteins (LDL) and triglycerides) was done per standard protocol.

### 2.7. Study Outcomes

Feasibility of the CCRT intervention was assessed by (a) percent of participants completing CCRT sessions among those allocated in the intervention and (b) percent of those allocated CCRT that completed the study protocol in a timely manner (8 weeks). Preliminary efficacy was assessed with respect to (a) performance-based neurocognitive tests (primary outcome), (b) QOL and frailty-related phenotype (FRP) (secondary outcomes). Outcome measures described below were evaluated twice—at study enrollment and again at week 6 following CCRT training or equivalent time period for the SOC group.

#### 2.7.1. Feasibility/Acceptability

Feasibility/acceptability of intervention was primarily evaluated as overall and HIV-status specific rate of study protocol completion (i.e., pre-intervention neuropsychological test, 20 CCRT sessions and post-intervention neuropsychological testing). A second measure of feasibility, timely completion, is defined to estimate number of adult participants that completed the study protocol within 8 weeks of enrolment.

#### 2.7.2. Cognitive Performance

Cognitive performance was assessed by performance on a set of standardized paper-and-pencil tests previously shown to be culturally-independent and sensitive for detecting HIV-associated neurocognitive disorder (HAND) and HIV dementia (HAD) in HIV+ patients in Uganda by members of our team [27]. All the tests have had their instructions and content translated into Luganda—the local language, which was used to present all test measures. Tests were scored according to manufacturer’s instructions and were analyzed as continuous outcome variables. The cognitive assessment battery included nine separate tests covering several cognitive domains, as described below:(a)Gross motor function—evaluated using Timed Gait test time to quantify amount of time (in seconds) taken to complete walking a 10-yard distance and back.(b)Fine motor function—evaluated using two tests: (i) Finger Tapping (number of taps within 10 s) and (ii) the Grooved Pegboard Test, which measures the amount of time (in seconds) it takes to place 25 slotted pegs into rows of randomly positioned slots on a board.(c)Executive function—measured using Color Trails2 test (time to completion of a complex task requiring attention to two aspects of problem solving simultaneously) and verbal (semantic) fluency (number correct animals named within 60 s) to quantify proficiency in completing tasks that require planning, organizing, prioritizing, focus/attention, multi-tasking, and inhibition of irrational impulses.(d)Speed of Processing—measures how long it takes individuals to complete a cognitive visual scanning and simple sequencing task using Color Trails 1 (time (in seconds) to task completion) and Digit Symbol test in which participants are asked to draw in small symbols in spaces under numbers corresponding to number/symbol pairs at the top of the test sheet (number correct matches within 90 s).(e)Attention/Short-term working memory—measured as number correct on Digit Span Forward and Digit Span Backward tests.(f)Verbal Learning/Memory—measures ability of individuals to acquire and retrieve components of memory using the WHO-UCLA Auditory Verbal Learning Test.

#### 2.7.3. Quality of Life

QOL was measured using the short form Medical Outcomes Study questionnaire. Individual scores for general health perceptions, physical functioning, pain, vitality, role functioning, social functioning, and mental health were computed. An overall QOL score was derived by summing these components and scaling total score to a theoretical maximum of 100.

#### 2.7.4. Frailty Related Phenotype

FRP is a clinical state of increased vulnerability resulting from aging associated decline in multiple physiologic systems. It was defined using the Edmonton frail scale to measure functional vulnerability in the following domains: cognition, general health status, functional independence, medication use, continence, functional performance, social support, and unintentional weight loss. An overall FRP score is the sum of respective domains and analyzed as a continuous outcome variable.

### 2.8. Statistical Issues

Statistical power: In light of the feasibility pilot nature of this intervention and absence of formal a priori hypotheses, specific power calculation was not applicable. Estimates of the effect sizes from the present study will inform power analyses for future more definitive investigation.

Statistical Analysis: Descriptive analyses by intervention arm and HIV status summarized the distributions of outcomes and potential covariates. To assess feasibility of CCRT intervention, the rates of protocol completion overall and according to HIV status were determined. Multivariable general linear models were used to estimate CCRT vs. SOC-related differences (β coefficients) in neurocognitive batteries, QOL, and FRP. The dependent variables in these models were primary/secondary outcome measures at week 8 (one at a time), and covariates included baseline version of the outcome, number of non-HIV comorbid conditions, age, sex, level of education, intervention group (CCRT vs. SOC), and HIV status. The coefficients for the intervention group variable (β) and their 95% confidence intervals (CIs) provided estimates of the intervention effects. In addition, effect sizes were calculated as differences between group means in the adjusted standard deviation units (square root of the mean squared error) to quantify clinical relevance of estimated βs and inform planning of future studies. Effect sizes of 0.5 or a larger are ubiquitously universal for definition of clinical importance, [28] and effect sizes of 0.33 or greater are often deemed clinically important as well [29]. All analyses were implemented in SAS v.9.4. To explore differences in CCRT effects according to HIV status; intervention group by HIV status interaction was added to the models; and estimates of the CCRT effects (β’s, 95% CIs, effect sizes) were obtained by HIV status.

### 2.9. Ethical Approval

This project was ethically reviewed and approved by the institutional review boards of Michigan State University, (Protocol #17-205) and Makerere University Research Ethics Committee (MUREC, protocol #: REC REF 0403-2017). In addition to these approvals, the implementation of this study protocol was further approved by the Uganda National Council for Science and Technology (Permit #: HS 2275).

## 3. Results

### 3.1. Description of Enrolled Participants

Male and female Ugandans between 50 and 87 years of age were enrolled. The sample included 54.6% female (*n* = 44) and by design equal number of HIV-infected (*n* = 40, mean age = 57.9 (SD (standard deviation) = 7) years) and age (±5 years), sex and village of residence-matched HIV-negative community controls (*n* = 40, mean age = 61.8 (SD = 7) years). All HIV+ adults currently received primary health care at Mildmay Hospital. Participants allocated CCRT vs. SOC had similar distributions of matching factors (age, sex, village of residence), as well years of education, prevalence of non-HIV comorbid diseases, duration of HIV disease, psychosocial stress, immune factors, and baseline performance in all neuropsychological test batteries (Table 1).

The median duration on cART among HIV+ adults was 10 years with a range of 1 to 18 years. Mean CD4 cell count was 564 (SD = 319) cells per microliter, and 57.5% (*n* = 23) were virologically suppressed. Lipid profile tests, educational level and prevalence of non-HIV comorbid conditions were similar across HIV groups. Average performance on WHO UCLA learning tests were similar across HIV groups. However, average performance in finger tapping, timed gait, score digit and grooved pegboard tests were higher for HIV-negative community controls relative to HIV+ older Ugandans (Table 2).

### 3.2. Feasibility/Acceptability of CCRT

Of the 41 individuals initially allocated CCRT, 38 (92.8%) completed the study protocol with 27 (71%) doing so within 8 weeks. Among the 21 HIV+ adults allocated CCRT, 19 (90.5%) completed the study protocol (Figure 1). Mean and median time to completion of CCRT intervention among HIV-infected adults were 7.9 (SD = 2.9) and 6.0 (range: 4.6 to 12.1) weeks respectively with 11 (57.9%) completing study protocol within 8 weeks of initial neuropsychological assessment. Among 20 demographically-matched HIV-negative adults allocated CCRT, all completed the study protocol with 17 (85%) of completions within 8 weeks. The mean and median times to completion of CCRT intervention among community controls were 6.4 (SD = 2.4) and 5.3 (range: 4.0 to 11.7) weeks respectively.

Overall Sample: Effect of CCRT on neurocognitive performance, QOL and frailty

Adjusted for baseline value of respective outcomes, non-HIV comorbid disease, HIV status, age, sex, and educational attainment; older adults that received CCRT had higher average post-intervention scores of clinical importance in tests verbal fluency (β = 1.25, 95%CI: −0.09 to 2.58, ES = 0.38); and the following subscales of the WHO UCLA tests of learning: interference list recall (β = 1.00, 95% CI: 0.02–1.98; ES = 0.43) and delayed recall (β = 1.04, 95%CI: 0.06–2.02; ES = 0.47) relative to peers allocated no CCRT intervention. However, time to completion of motor speed/hand-eye coordination tasks was modestly elevated for participants that received CCRT vs. no CCRT (grooved peg-board dominant hand: β = 0.21, 95%CI: −0.07, 0.50; ES = 0.32). All other CCRT-related differences in neurocognitive outcomes were statistically insignificant and of only small or modest clinical significance. There was no association between CCRT and post-intervention quality of life (β = −2.64, 95%CI: −6.7 to 1.4; ES = −0.28) and frailty-related phenotype (β = −0.12, 95%CI: −0.74 to 0.50; ES = −0.09) (Table 3).

### 3.3. Effect of CCRT on Neurocognitive Performance, QOL and Frailty according to HIV Status

Among HIV+ older adults, CCRT intervention resulted in post-intervention score improvement of large clinical importance in the areas of immediate recall (β = 0.91, 95%CI: 0.08–1.73; ES = 0.69), digit span backwards (β = 0.79, 95%CI: −0.18–1.77; ES = 0.51) and verbal fluency (β = 1.68, 95%CI: −0.32–3.67; ES = 0.51). Also among HIV+ older adults, the receipt of CCRT intervention resulted in higher scores of clinical importance in post-intervention delayed recall (ES: 0.41 to 0.44, *p* = 0.168 to 0.202) and finger tapping (ES = 0.38, *p* = 0.233) tests as well as faster time to completion of color trails (ES = −0.32, *p* = 0.306) and timed gait (ES = −0.44, *p* = 0.167) tasks.

Among HIV-negative demographically-matched controls, CCRT intervention on the one hand resulted in FRP declines of large clinical importance (β = −0.54, 95%CI: −1.00 to −0.08; ES = −0.71) and improvement in WHO UCLA learning tests for subscales of interference (ES = 0.45, *p* = 0.129) and delayed recall (ES = 0.45, *p* = 0.135) of clinical importance. On the other hand, receipt of CCRT intervention was associated with lower performance in digit span forward test (ES = −0.44, *p* = 0.147) and longer time to completion of grooved pegboard task (ES = 0.41, *p* = 0.172) among HIV negative community controls that received CCRT vs. no intervention (Table 4).

## 4. Discussion

With more than 90% of individuals assigned to CCRT completing the study protocol, and over 70% doing so within 8 weeks of enrolment, we concluded that the CCRT intervention is feasible among Ugandan adults 50 years and above. Although overall completion rate was similar by HIV status, timely protocol completion was lower among HIV-infected adults suggesting that future studies will need to increase efforts to encourage timely return for cognitive training among HIV-infected adults or potentially develop more local administration options. Overall findings suggest the potential utility of CCRT for mitigating HIV-and aging-related neurocognitive impairment in Ugandan adults 50 years and above. Given our sample size and the feasibility intent of this investigation, we interpret potential clinical relevance using a combination of statistical significance and differences in ES of ≥|0.33|, which corresponds to the difference of at least moderate clinical importance as per literature on patient reported outcomes [30,31,32]. The potential efficacy of CCRT-intervention reported here is similar to previous observations among patients with early Alzheimer’s Disease [33].

Study results show that regardless of HIV status, older adults who participated in CCRT showed improvements of at least modest clinical importance in tests of executive function (verbal fluency) and verbal learning/memory (interference list and delayed recall) relative to the control group. These preliminary findings of neurocognitive benefits of modest clinical importance in some cognitive domains are consistent with previous reports of CCRT-related neurocognitive benefit in healthy older individuals [34], older adults with HIV infection [35,36,37,38], and older individuals with mild cognitive impairment [39]. Specifically, among healthy older Americans (mean age = 69, SD = 6.9), CCRT targeting auditory perception and visuomotor/working memory domains of cognition was associated with post-training improvements in tests of everyday problem solving, working memory and matrix reasoning [34]. Our observations that CCRT improved post interference and delayed recall in older Ugandans is similar to previously reported cognitive training-related improvement in episodic recall and recognition in HIV-negative French individuals with mild cognitive impairment [39], and with improvement in tests of processing speed and time to completion of instrumental activities of daily living, in a small sample of HIV-infected American adults 40 years or older [35]. Hence, emerging data from the present study and others, including a recent meta-analysis of 12 studies in patients with mild cognitive impairment or subjective cognitive decline [40], are in support of the thesis that CCRT in populations at high risk of aging- and HIV-related neurocognitive dysfunction stimulates neuroplasticity that can build new and stronger connections between neurons and translate to measurable improvements observed in some neurocognitive domains [41]. However, not all studies have found CCRT-related neurocognitive benefits, including a previously reported observation of no cognitive performance benefit of a computer-based cognitive exercise program on age-related cognitive decline among American seniors [42].

We found more consistent and broader neurocognitive benefit of CCRT vs. no CCRT training among HIV-positive older Ugandans with clinically moderate or large cognitively beneficial effects noted in the areas of verbal learning/memory (immediate and delayed recall tests), short-term working memory (digit span backwards), executive function (verbal fluency), psychomotor function (finger tapping, timed gait), and processing speed (i.e., color trails). Of note, at enrolment, HIV-infected older adults had a broader scope of neurocognitive disadvantage relative to demographically-matched HIV-negative community controls with worse performance in motor function (Timed gait, finger tapping, dominant hand grooved pegboard), simple attention (digit span forward) and short-term verbal memory (digit span backward) tests. Hence, a larger benefit of CCRT intervention in HIV-infected older adult partly reflects the greater burden of neurocognitive impairment (and thus opportunity for CCRT-related remediation) in this group [43,44]. This position is supported by Kaur et al.’s observation that the largest post CCRT neurocognitive benefit in their sample occurred in HIV-positive adults that had the lowest baseline neurocognitive performance, higher viral load, poorer medication adherence, and the greatest number of years lived with HIV infection [45]. In spite of these hypotheses confirming association between CCRT and neurocognitive function among HIV-infected adults, in contrast with our study hypothesis, there was no evidence that CCRT intervention was beneficial for quality of life or FRP reduction among older adults living with HIV.

Among HIV-negative older adults, a large hypothesis confirming CCRT-related decline in FRP was observed although CCRT was unrelated to improvement in QOL. In fact, CCRT intervention-related neurocognitive benefit among HIV-negative older adults was relatively inconsistent and occurred in fewer cognitive domains compared to HIV-infected peers. The CCRT related reduction in frailty aligns with the observation that cognitive training in older patients with mild cognitive impairment was associated with modest improvements in self-rated QOL, activities of daily living and functional ability [46]. With respect to post-intervention cognitive scores, the CCRT-related moderate improvement observed in learning and memory tests among HIV-negative community controls was inconsistent with moderately worse scores in tests of simple attention and worse performance in test of hand-eye coordination/manual dexterity (grooved pegboard) observed in the same older adults that received CCRT vs. SOC. Future studies with larger sample size will be needed to clarify these findings.

To our knowledge, this is the first evaluation of the feasibility and promise of CCRT for mitigating aging and HIV-associated neurocognitive decline and for reducing FRP in Ugandan adults over 50 years old. Cultural adaptation of the CCRT intervention, utilization of open-source technology platform, and inclusion of HIV-positive and HIV-negative older Ugandans matched on key demographic factors are important strengths in this feasibility study. The fact that CCRT and SOC arms were similar at baseline with respect to several demographic covariates and pre-test neuropsychological performance, and attrition was low by study end, are additional strengths. However, this study is subject to the three limitations that should be considered in the interpretation of this study and in the design of definitive future studies: (a) CCRT intervention was not randomly allocated, limiting our ability to exclude confounding by unmeasured covariate; (b) post-test intervention duration was relatively short, excluding the ability to test for stability of observed associations, and (c) repeat neuropsychological testing did not use a parallel version of respective tools, raising the potential for learning effect upon repeat testing. These results are promising and suggests larger investigations are warranted to formally test the efficacy of CCRT for mitigation of premature neurocognitive decline, high risk of frailty and low quality of life in this vulnerable population.

## 5. Conclusions

Preliminary data from this feasibility study suggest that CCRT may be a promising non-pharmacologic intervention for mitigating neurocognitive decline and aging-related disability in older Africans, especially those with HIV infection. These findings underscore the need for prospectively powered clinical trials to test the efficacy of this intervention and the corresponding duration of any CCRT-related neuroprotective effects. If these preliminary findings are supported by more definitive studies, then CCRT represents a potentially effective yet affordable strategy for making a community-wide impact on a growing public health problem in SSA and beyond.

## Figures and Tables

**Figure 1 jcm-09-02137-f001:**
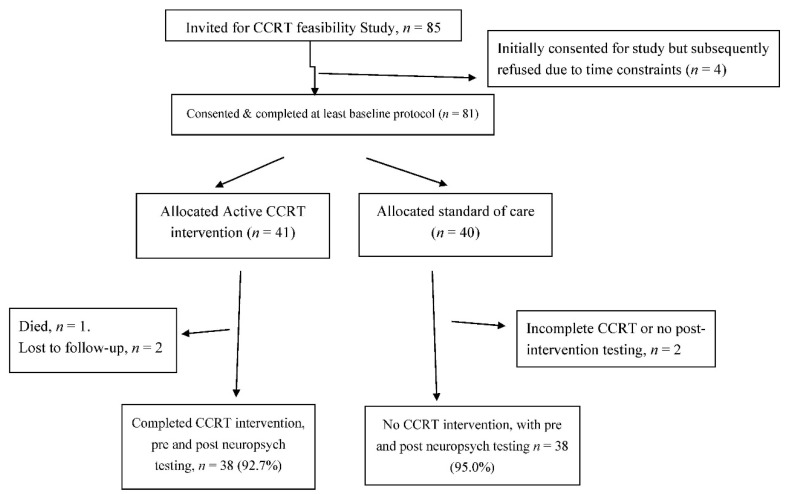
Sample selection to evaluate the feasibility and preliminary impact of Computerized Cognitive Rehabilitation Training on Performance Based Neurocognitive Function, Quality of Life and Frailty among Ugandan Adults 50 years and above, July 2017–October 2018. CCRT, computerized cognitive rehabilitation therapy.

**Table 1 jcm-09-02137-t001:** HIV-infected and demographically matched HIV-negative community dwelling adults 50–87 years old from the Wakiso District of Uganda non-randomly allocated CCRT vs. No CCRT for evaluation of intervention feasibility.

	CCRT, *n* = 41	No CCRT, *n* = 40	*p*-Value
Age (years)	59.7 (8.2)	60.0 (6.6)	0.894
	*n* (%)	*n* (%)	
Female Sex	24 (58.5)	20 (50.0)	0.441
HIV Status			
HIV+	19 (47.5)	20 (50.0)	0.823
Mean (SD) Years on HAART (if HIV+)	8.8 (5.6)	9.7 (3.8)	0.564
Years of Education			
≤Primary	21 (55.3)	17 (44.7)	0.539
Some/Completed Ordinary Levels	10 (26.3)	14 (36.8)	
Some Advanced Levels or Higher	6 (15.8)	6 (15.8)	
Number of non-HIV comorbid diagnoses			
0	11 (27.5)	13 (33.3)	0.838
1	18 (45.0)	16 (41.0)	
2	6 (15.0)	7 (18.0)	
3+	5 (12.5)	3 (7.7)	
Lipid Profile Tests	Mean (SD)	Mean (SD)	
High Cholesterol	16 (40.0)	14 (35.0)	0.559
Low HDL	16 (40.0)	14 (35.0)	0.559
High LDL	19 (47.5)	15 (37.5)	0.534
High Triglycerides	10 (25.0)	8 (20.0)	0.556
High CRP	2 (5.0)	7 (17.5)	0.127
CESD Score	16.7 (11.0)	16.1 (9.5)	0.816
Frailty Score	3.1 (1.9)	2.7 (1.5)	0.273
Quality of Life Score	72.1 (11.1)	71.0 (9.4)	0.621
Body Mass Index	24.4 (4.7)	23.6 (6.0)	0.562
Hemoglobin	14.3 (2.4)	14.2 (1.9)	0.746
Stress Scores	mean (SD)	mean (SD)	
Acute Stress	20.5 (6.2)	18.9 (7.0)	0.311
Recent Life Stress (within 5 years)	11.8 (5.6)	11.3 (4.9)	0.678
Number of Lifetime Adversity	5.5 (3.8)	5.6 (3.3)	0.963
Immune Measures	*n* = 34, mean (SD)	*n* = 34, mean (SD)	
Absolute CD4 cell count	612.8 (323)	663 (320)	0.516
CD4/CD8 Ratio	1.2 (0.6)	1.3 (0.9)	0.418
Neuropsychological Test Battery	Mean (SD)	Mean (SD)	
1. Timed Gait (time, s)	12.8 (7.6)	12.4 (4.2)	0.819
2. Verbal Fluency (Total Correct)	12.5 (2.9)	12.3 (3.4)	0.779
3. Symbol Digit Modality (number correct)	23.3 (12.3)	22.8 (11.7)	0.868
4. Finger Tapping (number of taps)	32.9 (7.4)	33.6 (9.3)	0.707
5. Overall (Forward + Backward)	8.5 (3.3)	7.3 (2.8)	0.108
6. Color Trails [100 × (T2−T1)/t1]	52.5 (54.4)	51.8 (46.6)	0.949
7. Grooved Pegboard Test (time, s)			
Non-Dominant Hand	150.9 (55.7)	146.0 (49.7)	0.677
Dominant Hand	123.2 (47.8)	119.5 (48.3)	0.739
8. WHO UCLA Verbal Learning Test (number correct)			
Sum of Immediate Recalls (Trials 1–5)	8.4 (1.7)	7.9 (1.8)	0.291
Interference List Recall (Trial 6)	5.0 (1.9)	4.6 (2.2)	0.335
Post Interference List Recall (Trial 7)	8.0 (2.5)	7.7 (2.5)	0.563
Delayed Recall of First list (Trial 8)	7.8 (2.6)	7.9 (2.7)	0.864
Delayed Recall & Recognition (Trial 9)	12.9 (2.1)	12.6 (2.7)	0.530

CRP = C-reactive protein, CCRT = computerized cognitive rehabilitation therapy, HIV = human immunodeficiency virus, HDL = high density lipoprotein, LDL = low density lipoprotein, SD = standard deviation, CD=cluster of differentiation.

**Table 2 jcm-09-02137-t002:** HIV-infected and demographically matched HIV-negative community dwelling adults 50–87 years old from the Wakiso District of Uganda non-randomly allocated CCRT vs. No CCRT for evaluation of intervention feasibility.

	HIV+, *n* = 40	HIV−, *n* = 41	*p*-Value
Age (years)	58.1 (7.3)	61.6 (7.3)	0.033
	*n* (%)	*n* (%)	
Female Sex	24 (58.5)	20 (50.0)	0.823
median (range) Years on HAART (if HIV+)	10 (1 to 18)	*n*/a	
Years of Education			
≤Primary	21 (55.3)	21 (52.5)	0.501
Some/Completed O’Levels	10 (26.3)	14 (35.0)	
Some A-Levels or Higher	8 (20)	5 (12.5)	
Number of non-HIV comorbid diagnoses	1.05 (1.5)	1.15 (1.6)	0.658
Lipid Profile Tests	Mean (SD)	Mean (SD)	
Cholesterol	5.0(1.13)	4.90 (0.99)	
HDL	1.40 (0.40)	1.40 (0.50)	0.929
LDL	2.92 (1.01)	3.12 (1.04)	0.403
Triglycerides	1.66 (1.3)	1.35 (0.46)	0.192
CRP	6562 (16464)	3310 (5068)	0.267
CESD Score	15.1 (11.5)	17.7 (8.8)	0.430
Frailty Score	3.1 (1.9)	2.9 (1.5)	0.501
Quality of Life Score	74 (11.0)	69.0 (9.3)	0.045
Body Mass Index	23.5 (5.1)	24.6 (5.7)	0.379
Hemoglobin	13.8 (1.6)	14.8 (2.5)	0.0376
Stress Scores	Mean (SD)	Mean (SD)	
Acute Stress	18.1 (6.4)	21.2 (6.6)	0.039
Recent Life Stress (within 5 years)	11.7 (6.2)	11.3 (4.1)	0.794
Number of Lifetime Adversity	5.5 (3.6)	5.6 (3.6)	0.963
Immune Measures	*n* = 34,	*n* = 34,	
	Mean (SD)	Mean (SD)	
Absolute CD4 cell count	564 (319)	722 (304)	0.042
CD4/CD8 Ratio	0.85 (0.38)	1.75 (0.80)	<0.001
Neuropsychological Test Battery	Mean (SD)	Mean (SD)	
1. Timed Gait (time, s)	13.9 (7.8)	11.4 (3.3)	0.075
2. Verbal Fluency (Total Correct)	12.2(3.1)	12.9 (2.5)	0.365
3. Symbol Digit Modality (number correct)	22.8 (11.6)	23.2 (12.4)	0.867
4. Finger Tapping (number of taps)	30.7 (9.2)	35.7 (6.3)	0.005
5a. Score digit forward (simple attention)	3.7 (1.8)	4.5 (1.9)	0.054
5b. Score digit backward (short term verbal memory)	3.4 (1.4)	4.1 (1.9)	0.064
5. Overall (Forward + Backward)	7.1 (2.9)	8.6 (3.2)	0.034
6. Color Trails [100 × (T2 − T1)/t1]	107.3 (68.5)	84.0 (56)	0.100
7. Grooved Pegboard Test (time, s)			
Non-Dominant Hand	0.17 (1.04)	−0.17 (0.94)	0.128
Dominant Hand	0.21 (1.1)	−0.21 (0.85)	0.060
8. WHO UCLA Verbal Learning Test (Number Correct)			
Sum of Immediate Recalls (Trials 1–5)	8.0 (1.7)	8.3 (2.5)	0.438
Interference List Recall (Trial 6)	5.0 (1.9)	5.0 (1.7)	0.438
Post Interference List Recall (Trial 7)	7.4 (2.3)	8.1 (2.4)	0.158
Delayed Recall of First list (Trial 8)	7.5 (2.4)	8.3 (2.4)	0.198
Delayed Recall & Recognition (Trial 9)	12.7 (2.4)	12.9 (2.5)	0.689

CRP = C-reactive protein, CCRT = computerized cognitive rehabilitation therapy, HIV = human immunodeficiency virus, HDL = high density lipoprotein, LDL = low density lipoprotein, SD = standard deviation, CD=cluster of differentiation.

**Table 3 jcm-09-02137-t003:** Age, sex, HIV status, and education-adjusted effect of CCRT vs. no CCRT intervention on performance in respective Neurocognitive Test Batteries.

	LS Means (SE)		β (95% CI)	Effect Size	*p*-Value	Adj.R^2^
	CCRT	No CCRT				
WHO UCLA Learning Tests (# Correct)						
Immediate Recall (trials 1–5)	9.94 (0.20)	9.60 (0.20)	0.33 (−0.29, 0.86)	0.25	0.243	0.46
Interference List Recall (trial 6)	5.44 (0.28)	5.72 (0.27)	−0.28 (−1.02, 0.46)	−0.16	0.407	0.25
Post Interference List Recall (trial 7)	10.1 (0.35)	9.1 (0.35)	**1.00 (0.02, 1.98)**	**0.43**	0.046	0.30
Delayed Recall (trial 8)	10.2 (0.36)	9.2 (0.35)	**1.04 (0.06, 2.02)**	**0.47**	0.038	0.41
Word Recognition (trial 9)	13.7 (0.32)	13.6 (0.32)	0.09 (−0.50, 0.99)	0.04	0.841	0.12
Digit Span Test (# correct)						
Short term auditory memory/simple attention	4.5 (0.19)	4.8 (0.19)	−0.36 (−0.88, 0.17)	−0.29	0.186	0.34
Short-term verbal memory	4.4 (0.24)	4.01 (0.25)	0.45 (−0.21, 1.12)	0.29	0.116	0.18
Overall score	8.76 (0.35)	8.72 (0.34)	0.03 (−0.91, 0.97)	0.01	0.949	0.34
Color Trails (time, s)	99.5 (9.7)	112.9 (9.7)	−13.4 (−39.8, 14.0)	−0.21	0.327	0.05
Finger Tapping Test (time, s)	38.1 (0.85)	37.3 (0.87)	0.83 (−1.6, 3.2)	0.15	0.498	0.05
Grooved Peg Board (time, s)						
Grooved Pegboard NDH	0.06 (0.11)	−0.12 (0.12)	0.18 (−0.16, 0.51)	0.24	0.296	0.45
Grooved Pegboard DH	0.07 (0.10)	−0.14 (0.11)	**0.21 (−0.07, 0.50)**	**0.32**	0.143	0.51
Verbal Fluency						
Verbal Fluency (Tot. Correct)	13.4 (0.85)	12.2 (0.91)	**1.25 (−0.09, 2.58)**	**0.38**	0.068	0.21
Timed Gait (time, s)	11.4 (0.52)	12.1 (0.52)	−0.65 (−2.1, 0.83)	−0.20	0.401	0.15
Symbol Digit Modality (# correct)	24.5 (1.9)	26.3 2.0)	−1.78 (−4.74, 1.19)	−0.25	0.239	0.63
Frailty score	1.98 (0.35)	2.09 (0.38)	−0.12 (−0.74, 0.50)	−0.09	0.714	0.22
Quality of Life Score	72.1 (2.6)	74.7 (2.8)	−2.64 (−6.7, 1.4)	−0.28	0.206	0.50

Estimates shown are age, sex, education and baseline score adjusted for each outcome measure in separate linear regression models. Measures of association (effect size and *p*-value for test of mean differences by intervention group) are shown for overall sample (regardless of HIV-status). Statistically significant differences and differences with corresponding effect size >|0.30| are bolded. Given the small sample size, we interpret potential clinical relevance using a combination of statistical significance and differences in ES of |0.3| which corresponds to at least a moderate effect per literature on patient-reported outcomes. ES differences of ≤|0.2| are considered ‘small’. # = number.

**Table 4 jcm-09-02137-t004:** CCRT Intervention-Related Change in Cognitive and QOL outcomes Among Older Ugandans with and without HIV infection.

	Among HIV-Positive Older Ugandans				Among HIV-Negative Older Ugandans			
	CCRT	No CCRT	Estimated Effect		CCRT	No CCRT	Estimated Effect	
	LS Means (SE)	LS Means (SE)	β (95% CI)	Effect Size (*p*-Value)	LS Means (SE)	LS Means (SE)	β (95% CI)	Effect Size (*p*-Value)
WHO UCLA Learning Tests (# Correct)								
Immediate Recall (trials 1–5)	10.1 (0.39)	9.2 (0.43)	**0.91 (0.08, 1.73)**	0.69 (0.032)	9.83 (0.42)	9.9 (0.43)	−0.19 (−0.96, 0.57)	−0.14 (0.620)
Interference List Recall (trial 6)	5.3 (0.52)	5.3 (0.59)	0.01 (−1.12, 1.13)	0.02 (0.992)	5.6 (0.57)	6.1 (0.59)	−0.50 (−1.52, 0.52)	−0.29 (0.334)
Post Interference List Recall (trial 7)	10.5 (0.70)	9.5 (0.77)	**1.03 (−0.43, 2.49)**	**0.44 (0.168)**	10.1 (0.75)	9.1 (0.76)	**1.05 (−0.31, 2.41)**	**0.45 (0.129)**
Delayed Recall (trial 8)	11.5 (0.68)	10.6 (0.75)	**0.91 (−0.49, 2.30)**	**0.41 (0.202)**	11.6 (0.73)	10.6 (0.39)	**1.00 (−0.31, 2.32)**	**0.45 (0.135)**
Word Recognition (trial 9)	13.4 (0.60)	13.7 (0.26)	−0.34 (−1.6, 0.92)	−0.15 (0.602)	14.0 (0.66)	13.6 (0.66)	0.46 (−0.74, 1.65)	0.20 (0.451)
Digit Span Test (# correct)								
Short term auditory memory/simple attention	5.05 (0.37)	5.3 (0.41)	−0.21 (−0.97, 0.56)	−0.17 (0.560)	4.7 (0.40)	5.3 (0.40)	−0.54 (−1.26, 0.19)	−0.44 (0.147)
Short-term verbal memory	4.35 (0.47)	3.56 (0.52)	**0.79 (−0.18, 1.77)**	**0.51 (0.112)**	4.47 (0.50)	4.26 (0.51)	0.20 (−0.72, 1.12)	0.13 (0.666)
Overall score	9.57 (0.65)	8.63 (0.73)	**0.95 (−0.42, 2.32)**	**0.32 (0.174)**	9.02 (0.71)	9.63 (0.71)	−0.62 (−1.92, 0.68)	−0.21(0.351)
Color Trails (time, s)	96.6 (19.0)	117 (21)	**−20.9 (−60.8, 19.1)**	**−0.32 (0.306)**	125 (20.4)	128 (20.7)	−3.2 (−40.1, 33.7)	−0.05 (0.867)
Finger Tapping Test (# taps)	38.2 (1.6)	36.2 (1.7)	**2.1 (−1.4, 5.6)**	**0.38 (0.233)**	37.7 (1.82)	37.5 (1.79)	0.2 (−3.1, 3.5)	0.04 (0.903)
Grooved Peg Board (time, s)								
Grooved Pegboard NDH	126.5 (10.7)	125 (11.6)	2.0 (−20.0, 24.0)	0.06 (0.860)	136(11.5)	129 (11.5)	6.6 (−14.2, 27.4)	0.20 (0.535)
Grooved Pegboard DH	97.9 (8.6)	92.5 (9.5)	5.5 (−12.2, 23.2)	0.20 (0.542)	102.9 (9.1)	91.6 (9.2)	**11.4 (−5.0, 27.7)**	**0.41 (0.172)**
Verbal Fluency								
Verbal Fluency (Tot. Correct)	13.2 (1.0)	11.5 (1.03)	**1.68 (−0.32, 3.67)**	**0.51 (0.099)**	13.5 (1.04)	12.7 (1.05)	0.87 (−0.99, 2.72)	0.26 (0.358)
Timed Gait (time, s)	9.5 (0.97)	10.9 (1.06)	**−1.43 (−3.5, 0.60)**	**−0.44 (0.167)**	9.9 (1.06)	9.8 (1.09)	0.12 (−1.8, 2.0)	0.04 (0.904)
Symbol Digit Modality(# correct)	23.6 (2.1)	26.0 (2.3)	**−2.4 (−6.8, 2.0)**	**−0.34 (0.286)**	25.5 (2.29)	26.7 (2.31)	−1.24 (−5.4, 2.9)	−0.17 (0.556)
Frailty score	2.24 (0.23)	2.22 (0.26)	0.02 (−0.47, 0.51)	0.03 (0.931)	2.29 (0.26)	2.83 (0.26)	**−0.54 (−1.00, −0.08)**	**−0.71 (0.023)**
Quality of Life Score	72.9 (3.2)	74.8 (3.2	−1.9 (−7.8, 4.1)	−0.20 (0.533)	71.1 (3.2)	73.2 (3.1)	−2.1 (−8.0, 3.9)	−0.22 (0.490)

Estimates shown are age, sex, education, and baseline score adjusted for each outcome (bolded) in separate linear regression models. Measures of association (effect size and p-value for test of mean differences by intervention group) are shown for overall sample (regardless of HIV-status) and within HIV-status groups. Within HIV groups, adjusted least-square means and SE for respective tests are shown for CCRT vs. no CCRT groups. Statistically significant differences and differences with corresponding effect size >|0.30| are bolded Given small sample size, we interpret potential clinical relevance using a combination of statistical significance and differences in ES of |0.30| which corresponds to at least moderate effect per literature on patient reported outcomes. ES differences of < = |0.2| are considered ‘small’. QOL = Quality of Life, # = Number.

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
