# Peer review of "Computerized Cognitive Rehabilitation Training for Ugandan Seniors Living with HIV: A Validation Study"

_jcm, 2020, doi:10.3390/jcm9072137_

Round 1
Reviewer 1 Report
The paper titled "Computerized cognitive rehabilitation training for Ugandan seniors living with HIV: a validation study" tackles an essential topic in the neuropsychology field, i.e. treatment of cognitive deficits via computerized training in senior citizens affected by HIV.
Authors' preliminary results are interesting from both clinical and research perspectives. The paper in its current format has significant merit, and only some minor points should be addressed before being accepted by the Journal of Clinical Medicine.
- Some recent papers have shown on a large basis that computerized training is an interesting therapeutic option for elderly individuals: I would strongly suggest the authors to include these references in their manuscript (Cavallo et al. 2016 published on Archives of Clinical Neuropsychology; Hu et al. 2019 published on J Neurol).
- Page 4 line 172: I would recommend to eliminate the word "objectively", as a neuropsychological assessment cannot allow us to get "objective" piece of information;
- Table 2 presents with some graphical problems (it is difficult to read the very last column);
- It would be interesting to know within each treatment group whether there was a difference in demographic variables (e.g., age and education) between sex, as we only know that roughly half of participants were males and half females, but we do not know distribution of demographic variables among gender;
- In the Discussion, the authors should briefly summarize all the limits this study presents: e.g., as reported in the Table 2, the two groups of HIV-infected and demographically matched HIV-negative community dwelling adults were significantly different in terms of age; in addition, parallel versions of neuropsychological tests were not used at the post-treatment, and thus the pattern of results might have been influenced by the "test-retest effect"; lastly, the absence of a follow-up (e.g., two months after the end of training) does not allow us to know anything about stability over time of the results achieved during training.
Provided that these minor points are rectified, in my view the manuscript will reach a very good level.
Author Response
Thank you for helpful review of our manuscript. Response to critique from all reviewers are included in the attached file.
Appreciatively,
Amara Ezeamama
The paper titled "Computerized cognitive rehabilitation training for Ugandan seniors living with HIV: a validation study" tackles an essential topic in the neuropsychology field, i.e. treatment of cognitive deficits via computerized training in senior citizens affected by HIV.
Authors' preliminary results are interesting from both clinical and research perspectives. The paper in its current format has significant merit, and only some minor points should be addressed before being accepted by the Journal of Clinical Medicine.
Response: We thank reviewer #1 for recognizing the merits of our study and providing critical comments to further improve this communication.
- Some recent papers have shown on a large basis that computerized training is an interesting therapeutic option for elderly individuals: I would strongly suggest the authors to include these references in their manuscript (Cavallo et al. 2016 published on Archives of Clinical Neuropsychology;Hu et al. 2019 published on J Neurol).
Response: Thank you for bringing these two studies to our attention. Both were deemed appropriate and are now cited in the discussion section.
- Page 4 line 172: I would recommend to eliminate the word "objectively", as a neuropsychological assessment cannot allow us to get "objective" piece of information;
Response: We have now eliminated this word as suggested. Thank you.
- Table 2 presents with some graphical problems (it is difficult to read the very last column);
Response: This table has now been replaced with another without the identified problem.
- It would be interesting to know within each treatment group whether there was a difference in demographic variables (e.g., age and education) between sex, as we only know that roughly half of participants were males and half females, but we do not know distribution of demographic variables among gender;
Response: As shown in Table 1, by CCRT arm, there was no di difference among participants by age, sex distribution, percent HIV+, years of education and number of comorbid non-HIV chronic disease.
Within CCRT arms, females were similar to males with respect to HIV-prevalence, years of HAART treatment and education. Adults not allocated CCRT were comparable by sex with respect to age. Among adults allocated CCRT, women were slightly younger than men. Because of the small sample size within columns with the double stratification by treatment arm and sex, this information is not integrated into the manuscript but provided below for the reviewers’ information.
|
|
Allocated CCRT |
No CCRT |
||||
|
|
Male |
Female |
P-value |
Male |
Female |
P-value |
|
Age (years) |
61.5 (10.1) |
58.4 (4.9) |
0.0452 |
61.0 (6.9) |
59.0 (6.6) |
0.3157 |
|
Yrs HAART (if HIV+) |
8.12 (5.5) |
9.5 (5.9) |
0.6377 |
10.3 (3.3) |
9.1 (4.1) |
0.4816 |
|
%HIV+ |
17 (52.9) |
24 (45.8) |
0.6634 |
21(52.4%) |
20 (0.50) |
0.988 |
|
Education <=Primary Some /Completed O’levels Some A’Levels |
10 (66.7) 2 (13.3) 3 (20.0) |
11 (50.0) 8 (36.4) 3 (13.6) |
0.300
|
6 (28.6) 11 (52.4) 4 (19.0) |
11 (61.1) 4 (22.2) 33 (16.7) |
0.091 |
- In the Discussion, the authors should briefly summarize all the limits this study presents: e.g., as reported in the Table 2, the two groups of HIV-infected and demographically matched HIV-negative community dwelling adults were significantly different in terms of age; in addition, parallel versions of neuropsychological tests were not used at the post-treatment, and thus the pattern of results might have been influenced by the "test-retest effect"; lastly, the absence of a follow-up (e.g., two months after the end of training) does not allow us to know anything about stability over time of the results achieved during training.
Response: In light of the above, the following narrative has been added to the revised manuscript. “However, this study is subject to the three limitations that should be considered in the interpretation of this study and in the design of definitive future studies: a) CCRT intervention was not randomly allocated limiting our ability to exclude confounding by unmeasured covariate; b) post-test intervention duration was relatively short excluding the ability to test for stability of observed associations and c) repeat neuropsychological testing did not use parallel version of respective tools raising the potential for learning effect upon repeat testing.”
Provided that these minor points are rectified, in my view the manuscript will reach a very good level.
Reviewer 2 Report
The study aims to assess the feasibility and the preliminary efficacy of computerized cognitive rehabilitation therapy (CCRT) vs. a standard of care (SOC) in a sample of Ugandan seniors living with or without HIV. The research is conducted with two not randomized arms, the CCRT and SOC, and compares the effects of the CCRT on HIV-positive and HIV-negative older Ugandans. Further variables (i.e. age, sex, level of education, etc.) were also taken into account and used as covariates into multivariate regression models.
The issue is relevant for different purposes: above all, to understand how to intervene to mitigate cognitive and psychomotor impairments and to improve the quality of life of Ugandan seniors with HIV.
The study is well introduced, and the paper is well written. The tables are usually clear and easy to read. I noticed only an impediment in table 2, due probably to the conversion to the PDF format: the p values are not easily readable because the right margin appears truncated. The results are well discussed, and the conclusions are supported by the findings obtained.
However, I found some major flaws that I’ll summarize in the following remarks.
- The procedure used to allocate participants to the two arms can have introduced a bias in the results because the first eight participants who adhere to CCRT could be also more motivated and compliant about the study, compared to the following eight.
- Considering that at enrolment HIV-infected seniors showed worse performance in motor function, in simple attention, and short-term verbal memory, it cannot be excluded that their improvement has been due to regression towards the mean or a slower impairment compared to the HIV-negative group, that show a significantly higher mean age.
- Concerning post-intervention cognitive scores, the moderate improvement observed in learning and memory tests compared to moderately worse scores observed in simple attention and psychomotor functions raise questions about some confounding variables that have not been sufficiently controlled, such as i.e. physical activity levels.
I think the authors must address these issues or report them as a limit of their study.
Minor flaws
Although the attrition rate was negligible, it could be interesting to report how affected positive vs. negative HIV participants.
In table 4 the sign of the effect size of the Quality of Life Score among HIV-positive should be negative.
Line 325 replace “higher” to “lower”
Line 342 replace “matric” with “matrix”
Line 358 replace “rails” with “trails”
Author Response
Thank you for helpful review of our manuscript. Response to critique from all reviewers are included in the attached file.
Appreciatively,
Amara Ezeamama
The study aims to assess the feasibility and the preliminary efficacy of computerized cognitive rehabilitation therapy (CCRT) vs. a standard of care (SOC) in a sample of Ugandan seniors living with or without HIV. The research is conducted with two not randomized arms, the CCRT and SOC, and compares the effects of the CCRT on HIV-positive and HIV-negative older Ugandans. Further variables (i.e. age, sex, level of education, etc.) were also taken into account and used as covariates into multivariate regression models.
The issue is relevant for different purposes: above all, to understand how to intervene to mitigate cognitive and psychomotor impairments and to improve the quality of life of Ugandan seniors with HIV.
The study is well introduced, and the paper is well written. The tables are usually clear and easy to read. I noticed only an impediment in table 2, due probably to the conversion to the PDF format: the p values are not easily readable because the right margin appears truncated. The results are well discussed, and the conclusions are supported by the findings obtained.
Response: We thank the reviewer for appreciating the contribution of this study. Table 2 has now been replaced with another without the identified problem.
However, I found some major flaws that I’ll summarize in the following remarks.
- The procedure used to allocate participants to the two arms can have introduced a bias in the results because the first eight participants who adhere to CCRT could be also more motivated and compliant about the study, compared to the following eight.
Response: We agree that non-randomized allocation could introduce bias. This has prominently been discussed as a limitation. “However, this study is subject to the three limitations that should be considered in the interpretation of this study and in the design of definitive future studies: a) CCRT intervention was not randomly allocated limiting our ability to exclude confounding by unmeasured covariate; b) post-test intervention duration was relatively short excluding the ability to test for stability of observed associations and c) repeat neuropsychological testing did not use parallel version of respective tools raising the potential for learning effect upon repeat testing.”
- Considering that at enrolment HIV-infected seniors showed worse performance in motor function, in simple attention, and short-term verbal memory, it cannot be excluded that their improvement has been due to regression towards the mean or a slower impairment compared to the HIV-negative group, that show a significantly higher mean age.
Response: We note that equal number of HIV-infected seniors were included in each arm. To the extent that this occurred, it is likely to do so equally by intervention arm. Therefore, CCRT related improvement in these measures are unlikely to be driven by existing differences at baseline.
- Concerning post-intervention cognitive scores, the moderate improvement observed in learning and memory tests compared to moderately worse scores observed in simple attention and psychomotor functions raise questions about some confounding variables that have not been sufficiently controlled, such as i.e. physical activity levels.
Response: We agree that residual confounding due to unmeasured covariates can not be excluded and have included this as a limitation of our study as in #1 above.
I think the authors must address these issues or report them as a limit of their study.
Response: We hope you would find our specific actions noted above to be sufficiently responsive to your critiques. Thank you.
Minor flaws
Although the attrition rate was negligible, it could be interesting to report how affected positive vs. negative HIV participants.
In table 4 the sign of the effect size of the Quality of Life Score among HIV-positive should be negative.
Line 325 replace “higher” to “lower”
Response: In response to the above, we checked the identified location and all sentences including “higher”. We did not find any of them to be inappropriate.
Line 342 replace “matric” with “matrix”. Response: Done. Thank you.
Line 358 replace “rails” with “trails” Response: Done. Thank you.
Reviewer 3 Report
The study is well conducted and well articulated. I would like to make minor suggestions.
I understand that English and Swahili are official languages in Uganda. I imagine that the CCRT was administered through English, but this is not clear.
The title contains the word "validation". I understand the intent to validate western findings in Ugandan population, but it may help readers if you can mention the differences that *could* have prevented validation.
The study does not require the results to be presented graphically. However, it may help readers to catch this study, if you have the main results presented graphically (by a graph), it will be easier for this study to sell. I don't mean to require this, but just to suggest so that this study receives attention it deserves.
Author Response
Thank you for helpful review of our manuscript. Response to critique from all reviewers are included in the attached file.
Appreciatively,
Amara Ezeamama
The study is well conducted and well articulated. I would like to make minor suggestions.
I understand that English and Swahili are official languages in Uganda. I imagine that the CCRT was administered through English, but this is not clear.
Response: Official languages in Uganda are English and Luganda. In the methods section, the following is included to show that instructions and directions for the CCRT intervention were provided in the local language. “Research assistants proficient in the local language, Luganda, supervised CCRT and provided instructions or guidance as needed.”
The title contains the word "validation". I understand the intent to validate western findings in Ugandan population, but it may help readers if you can mention the differences that *could* have prevented validation.
Response: We appreciate this critique. We have now included a prominent set of limitation narrative as follows: “However, this study is subject to the three limitations that should be considered in the interpretation of this study and in the design of definitive future studies: a) CCRT intervention was not randomly allocated limiting our ability to exclude confounding by unmeasured covariate; b) post-test intervention duration was relatively short excluding the ability to test for stability of observed associations and c) repeat neuropsychological testing did not use parallel version of respective tools raising the potential for learning effect upon repeat testing.”
The study does not require the results to be presented graphically. However, it may help readers to catch this study, if you have the main results presented graphically (by a graph), it will be easier for this study to sell. I don't mean to require this, but just to suggest so that this study receives attention it deserves.
Response: We carefully considered this suggestion and decided that presenting the results in tabular form provides more information to the invested reader and allows them anticipate potential power challenges within sub-groups.